# GENERALIZABLE HUMAN GAUSSIANS FROM SINGLE-VIEW IMAGE

**Jinnan Chen[1], Chen Li[1], Jianfeng Zhang[1], Lingting Zhu[2], Buzhen Huang[1],
Hanlin Chen[1], Gim Hee Lee[1]**
[1]National University of Singapore, [2]The University of Hong Kong
`jinnan.c@u.nus.edu, gimhee.lee@nus.edu.sg`

## ABSTRACT

In this work, we tackle the task of learning 3D human Gaussians from a single image, focusing on recovering detailed appearance and geometry including unobserved regions. We introduce a single-view generalizable Human Gaussian Model (HGM), which employs a novel generate-then-refine pipeline with the guidance from human body prior and diffusion prior. Our approach uses a ControlNet to refine rendered back-view images from coarse predicted human Gaussians, then uses the refined image along with the input image to reconstruct refined human Gaussians. To mitigate the potential generation of unrealistic human poses and shapes, we incorporate human priors from the SMPL-X model as a dual branch, propagating image features from the SMPL-X volume to the image Gaussians using sparse convolution and attention mechanisms. Given that the initial SMPL-X estimation might be inaccurate, we gradually refine it with our HGM model. We validate our approach on several publicly available datasets. Our method surpasses previous methods in both novel view synthesis and surface reconstruction. Our approach also exhibits strong generalization for cross-dataset evaluation and in-the-wild images. We open-source our code at: `https://github.com/jinnan-chen/HGM`.

## 1 INTRODUCTION

Automatic 3D human reconstruction from single image is crucial in augmented and virtual reality (AR/VR), game industry, filmmaking, *etc*. Previous works rely on strong 3D supervision such as the signed distance value or occupancy (Saito et al., 2019; 2020; Zhang et al., 2023c; Xiu et al., 2022; 2023; Zhang et al., 2024; Ho et al., 2024) and focus on surface reconstruction, neglecting novel view synthesis quality, resulting in smoothed and blurred textures. With the development of neural radiance fields (Mildenhall et al., 2020), novel view rendering quality has been greatly improved for human appearance modeling (Hu et al., 2023; Kwon et al., 2021; Gao et al., 2022). However, due to the ill-posed nature of single view reconstruction, the back and side views are always blurry and lack details without additional prior. Furthermore, these methods needs large amounts of query points sampled for volume rendering, which hinders practical real-time application in the industries. Some other methods optimize underlying appearance and geometry from scratch by introducing score distillation sampling during the optimizationTang et al. (2024b); Cao et al. (2024). Although effective, these methods still suffer from slow optimization and over-saturation problems. Recent 3D Gaussians generation methods (Tang et al., 2024a; Yinghao et al., 2024) combine multi-view diffusion models (Liu et al., 2023; Wang & Shi, 2023; Shi et al., 2023; Li et al., 2024; Xu et al., 2023) with generalizable multi-view Gaussians prediction models to generate 3D Gaussians with high quality and efficiency. We aim to extend this on human reconstruction. However, directly employ such methods to human reconstruction with complex texture and poses gives unsatisfying results due to: 1) Inconsistency across multiple views: The multi-view images generated from diffusion model lacks consistency in appearance and pose across different viewpoints. This inconsistency stems from the inherent complexity of human body structure and movement, which leads to low-quality reconstruction results. 2) Quality loss in front view reconstruction: multi-view diffusion process involves down-sampling and changing the original input image. This step results in significant quality degradation when reconstructing the front view image, compromising the fidelity to the original input. 3) Estimating SMPL-X parameters from single view input is ill-posed, directly applying initially

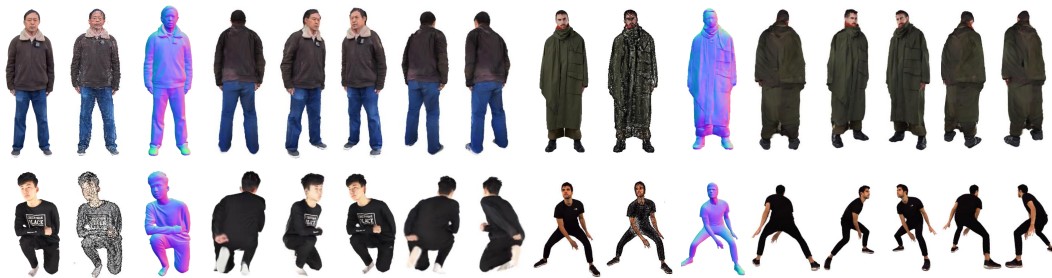

Figure 1: Our method reconstructs detailed and geometrically consistent human Gaussian models from single view images, including loosing clothes, challenging pose and in-the-wild images.

estimated SMPL-X can lead to bending legs and wrong elevation issues in previous method Xiu et al. (2022; 2023); Zhang et al. (2024); Ho et al. (2024).

To address the above-mentioned problems, we introduce a novel Human Gaussians Model (HGM), which supports fast and high quality rendering from single view input, and generalize well to loosing clothes, challenging poses and in-the-wild images as shown in Fig. 1. We do not use multi-view diffusion models due to the multiview inconsistency and resolution degradation problem. Instead, we propose a coarse-to-fine framework, where the diffusion model is adapted to refine back-view images rendered from our predicted coarse human Gaussians. In this way, we can keep the resolution and content of the original input image for high-fidelity reconstruction. In order to model the complex structure of a human, we inject the human prior into the Gaussian prediction process. Specifically, our model consists of two branches: 1) The first branch is a UNet to directly predict Gaussians from the input image, as inspired by image splatter (Szymanowicz et al., 2024). 2) The second branch uses learnable tokens attached to SMPL-X vertices for structural feature extraction with attention layers, and then combined with UNet features with SparseConv Graham et al. (2018) and a transformer for Gaussian enhancement. Recognizing the inaccurate estimate of SMPL-X from the pre-trained model, during inference, we iteratively refine the initial SMPL-X parameters with our HGM pre-trained with ground truth SMPL-X. Given the loss of details of the back view by directly predicting the Gaussians from a single view, we further apply a ControlNet to refine the back view with the control signal from the back-view image rendered from the coarse stage. We then input the original front view and refined back view images to our HGM model to get the final refined Gaussians. Meshes can be extracted from densely rendered depth map and TSDF fusion. Our model can be trained with only posed multiview images without 3D supervision and generalizes well to untrained datasets and in-the-wild images.

In summary, our contributions are:

- We introduce a generate-then-refine pipeline for single view human Gaussian reconstruction that leverages diffusion priors for back view refinement, avoiding the multi-view inconsistencies commonly observed in multiview diffusion models.

- Our proposed dual-branch reconstruction pipeline incorporates human priors by attaching learnable tokens to the SMPL-X vertices for structural feature extraction. We then fuse these features from the SMPL-X branch with the U-Net branch using Sparse Convolution and transformer.

- To address potential inaccuracies in initial SMPL-X estimations, we employ our Human Gaussian Model (HGM) to iteratively refine the estimated SMPL-X parameters, resulting in better alignment.

- Through extensive experimentation, we demonstrate the efficacy of our method in both novel view synthesis and 3D reconstruction tasks. Our approach consistently achieves state-of-the-art performance on various metrics and benchmarks.

## 2 RELATED WORKS

**Single-view Human Reconstruction.** PIFu (Saito et al., 2019), PIFuHD (Saito et al., 2020), PaMIR (Zheng et al., 2021), and GTA (Zhang et al., 2023c) are capable of inferring full textures from a single

image. Techniques such as PHORHUM (Alldieck et al., 2022) and S3F (Corona et al., 2023) go further by separating albedo and global illumination. However, these methods lack information from other views or prior knowledge, such as diffusion models, often resulting in unsatisfactory textures. TeCH (Huang et al., 2024) utilizes diffusion-based models to visualize unseen areas, producing realistic results. However, it requires time-intensive optimization per subject and is dependent on accurate SMPL-X. The emergence of Neural Radiance Fields (NeRF) has led to methods (Hu et al., 2023; Huang et al., 2023; Gao et al., 2022; Kwon et al., 2021) using videos or multi-view images to optimize NeRF for the capture of human forms. Recent advances such as SHERF (Hu et al., 2023) and ELICIT (Huang et al., 2023) aim to generate human NeRFs from single images. Although NeRF-based approaches are effective in creating high-quality images from various perspectives, they often struggle with detailed 3D mesh generation from single images and require extensive optimization time. More recently, SiTH (Ho et al., 2024) proposes to combine a back-view hallucination model with an SDF-based mesh reconstruction model. Similarly, SIFU (Zhang et al., 2024) employs a text-to-image diffusion-based prior to generating consistent textures for invisible views. However, these methods require 3D annotations such as the SDF of the meshes and texture maps as strong supervision and still fail to generate renderings with high fidelity due to the limited 3D training data and representation. In addition, these methods suffer from SMPL estimation errors, leading to bending legs and wrong elevation of the reconstructed 3D humans. Compared to these methods, our approach can be trained solely on multi-view images and achieves much better novel view synthesis quality. **Human Gaussians.** 3D Gaussians (Kerbl et al., 2023) and differentiable splatting (Szymanowicz et al., 2024) have gained broad popularity due to their efficiency in reconstructing high-fidelity 3D scenes from posed images using only a moderate number of 3D Gaussians. This representation has been quickly adopted for various applications, including imag or text-conditioned 3D generation and avatar reconstruction. Among these methods, Gauhuamn and HUGS (Hu & Liu, 2024; Kocabas et al., 2024) are the first to propose optimizing human Guassians from monocular human videos. However, they are not applicable to single static human images. GPS-Gaussian(Zheng et al., 2024) propose a generalizable multi-view huaman Gaussian model with high quality rendering; however, it needs dense views 16 or 8, which cannot be directly applied to single-view human images. Our human model achieves strong generalization in generating human Gaussians from single-view images, complementing concurrent work such as Pan et al. (2024).

**Generalizable Gaussians with Multi-view Diffusion.** The Large Reconstruction Model (LRM) (Hong et al., 2024) scales up both the model and the dataset to predict a neural radiance field (NeRF) from single-view images. Although LRM is primarily a reconstruction model, it can be combined with Diffusion Models (DMs) to achieve text-to-3D and image-to-3D generation as demonstrated by extensions such as Zero123(Liu et al., 2023), Image Dream(Wang & Shi, 2023) Instant3D (Li et al., 2024) and DMV3D (Xu et al., 2023). Our method also builds on a strong reconstruction model and uses pre-trained 2D DMs to provide input images missing information in a feedforward manner. Some concurrent works, such as LGM (Tang et al., 2024a), AGG (Xu et al., 2024), and Splatter Image (Szymanowicz et al., 2024), also utilize 3D Gaussians in a feed-forward model. LGM (Tang et al., 2024a) combines novel view generation diffusion models with generalizable Gaussians in a feedforward manner, while GRM (Yinghao et al., 2024) replaces the U-Net architecture with a pure-transformer one and scales up to large resolution. However, these methods face two main challenges when using pre-trained diffusion models. Firstly, the generated input view image becomes blurry compared to the original input, which affects the subsequent generalizable Gaussian model. Secondly, diffusion models can introduce multiview inconsistency, especially for human images with different poses, making direct adaptation unfeasible. We solve these problems by using ControlNet as the refinement tools without damaging the input image quality or introducing multi-view inconsistency.

## 3 OUR METHOD

### 3.1 PRELIMINARIES

**3D Gaussian Splatting (3DGS).** Introduced by (Kerbl et al., 2023), 3D Gaussian splatting represents 3D assets or scenes using a collection of 3D Gaussians. Each Gaussian is characterized by its center $x \in \mathbb{R}^3$, scaling factor $s \in \mathbb{R}^3$, rotation $r \in \mathbb{R}^3$, opacity $\alpha \in \mathbb{R}$, and color features $c \in \mathbb{R}^c$. View-dependent effects can be modeled with spherical harmonics. 3D scenes can be explicitly represented by a set of Gaussians $G = \{G_i\}$, where $G_i = \{x_i, s_i, r_i, \alpha_i, c_i\}$ represents the attributes for the $i$-th Gaussian. Compared with NeRF (Mildenhall et al., 2020), 3DGS performs fast rendering by

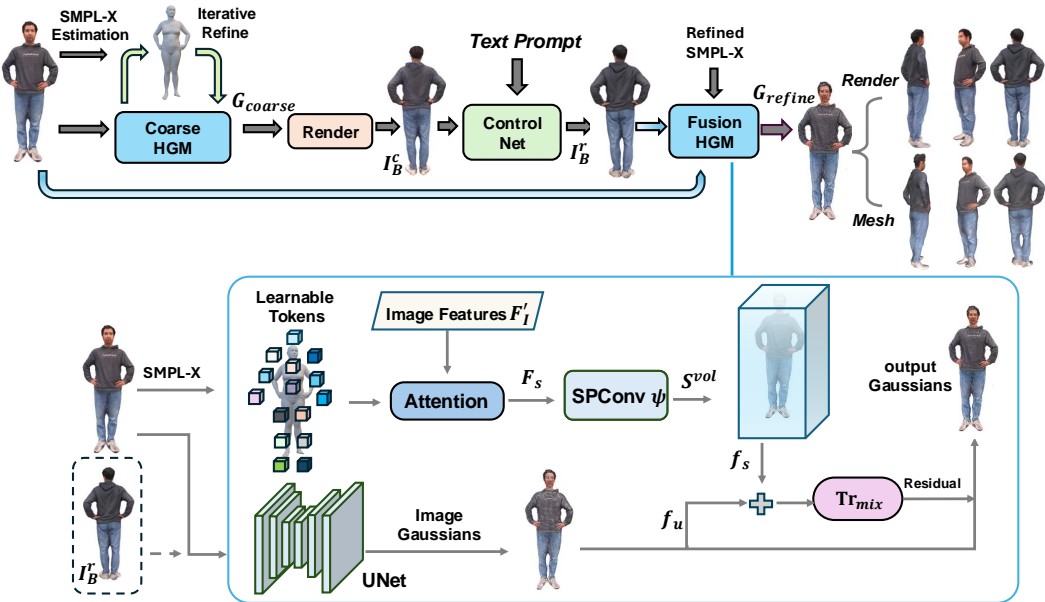

Figure 2: Our framework and HGM model. (Top) Our framework consists of three steps: 1) Coarse Gaussians prediction with iterative SMPL-X refinement. 2) Back view refinement with ControlNet. 3) Two view reconstruction to get the refined $G_{refine}$. (Bottom) Our HGM model consists of two branches: Image Gaussians prediction by UNet and adding additional structural features extracted from SMPL-X branch. $f_{smpl}$ are sampled by the Gaussian centers from the SMPL-X volume $S^{vol}$ and fused with $f_u$ to the fusion transformer $\text{Tr}_{mix}$ to obtain the Gaussian output.

first projecting Gaussians onto the image plane as 2D Gaussians and performing alpha-blending for each pixel in front-to-back depth order. Building on this, Image Splatter (Szymanowicz et al., 2024) proposes predicting Gaussians from a single image through image-to-image translation. Specifically, each pixel is converted to a Gaussian with corresponding attributes, supervised by multi-view images. Our model builds on this representation by directly predicting XYZ coordinates from the image instead of the depth.

## 3.2 OVERVIEW

Fig. 2 shows an overview of our framework. Given a single input human image $I$, our aim is to predict the corresponding human Gaussians, which can be further rendered for novel view synthesis and mesh extraction. As shown in the upper part of Fig. 2, our proposed method consists of three parts: 1) **Coarse Gaussians prediction with SMPL-X refinement (*cf.* Sec. 3.3)** 2) **Back-view refinement with ControNet (*cf.* Sec. 3.4).** 3) **Two-view reconstruction (*cf.* Sec. 3.5).**

## 3.3 COARSE GAUSSIANS PREDICTION WITH SMPL-X REFINEMENT

### 3.3.1 OUR HGM MODEL

The lower part of Fig. 2 shows our proposed Human Gaussian Model (HGM). The direct prediction of Gaussians from the image pixels with UNet (Szymanowicz et al., 2024; Tang et al., 2024a) lacks human shape and pose prior, thus leading to unsatisfactory results. We therefore introduce a dual branch that utilizes SMPL-X to enforce human shape and pose prior to the Gaussian prediction process. Specifically, for UNet branch, the collection of the RGB value and ray embedding for each pixel are concatenated into a 9-channel feature map as the input $F_I = \{c_i, o_i \times d_i, d_i \mid i = 1, 2, ..., N\}$. Our HGM model predicts Gaussians from the U-Net as:

$$G_u = \text{UNet}(F_I), \tag{1}$$

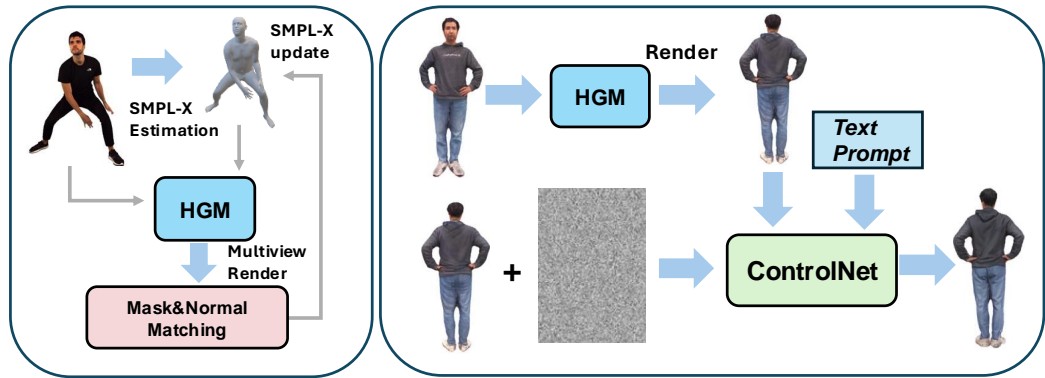

Figure 3: Left: Our SMPL-X refinement pipeline. Right: Our back-view refinement ControlNet.

which we refer to as Image Gaussians. For the SMPL-X branch, we attach learnable tokens to each of the SMPL-X vertices and extract the patch features of the image, denoted $F'_I$. We use cross-attention between these learnable tokens and the patch features to obtain $F_S$. This approach takes advantage of the fact that SMPL-X vertices are defined in semantically similar areas across different identities. Consequently, the learned tokens can memorize the mapping from the training dataset to unseen identities during inference, effectively providing structural human priors. This mechanism enables our model to capture and utilize semantic consistent features across diverse identities, enhancing its ability to generalize to new subjects. We apply SparseConvNet (Graham et al., 2018) (SPConv $\Psi$) to propagate the SMPL-X features to the predefined whole bounding box, and we denote this feature as the SMPL-X volume feature: $S^{vol} = \Psi(F_S)$, where $F_S$ are the SMPL-X features. The volume feature reconstructed from the SMPL-X vertex feature provides geometric cues of the target human body. The centers of the Image Gaussians from $G_u$ are then used to sample the propagated SMPL-X volume features, denoted as $f_s = S^{vol}(C_u)$, where $C_u$ are the centers of $G_u$. SMPL-X $f_s$ features are then concatenated with the features of UNet $f_u$ for each Gaussian. This concatenated feature is fed into a transformer $\text{Tr}_{mix}$ to obtain the coarse Gaussians, *i.e.*:

$$G_{coarse} = \text{Tr}_{mix}([f_u, f_s]). \tag{2}$$

Specifically, we predict the xyz coordinates residuals for the Image Gaussians and all the other updated Gaussian features. $\text{Tr}_{mix}$ is a transformer that contains multiple self-attention blocks among Gaussians to ensure that each Gaussian is aware of the other.

### 3.3.2 SMPL-X REFINEMENT

Given initial estimated SMPL-X is not accurate, we leverage our pre-trained HGM to iteratively refine the SMPL-X parameters based on the mask and optionally normal matching as shown in Fig. 3 left part. Specifically, we rendered the side-view masks and normals(optionally). We minimize the mask and normal difference between the Gaussian and SMPL-X renderings and back-propagate the loss to SMPL-X parameters. Then we interatively input the updated SMPL-X to our HGM model, so the Gaussian is also updated to give more accurate masks. For normal matching, we use the pre-trained normal estimator from Xiu et al. (2022) that also needs SMPL-X as input and iteratively update the SMPL-X parameters. Specifically, we compute:

$$\mathcal{L}_{SMPL-X} = \lambda_{front}\mathcal{L}_{front} + \lambda_{side}\mathcal{L}_{side} + \lambda_n\mathcal{L}_{normal}, \tag{3}$$

and back propagate it to the SMPL-X parameters. $\mathcal{L}_{normal}$, $\mathcal{L}_{front}$ and $\mathcal{L}_{side}$ are losses between the rendered masks and 2D detected keypoints of SMPL-X model and the coarse Gaussians by HGM. $\mathcal{L}_{normal}$ is the loss between the rendered SMPLX normal and the pre-trained normal estimator (Xiu et al., 2022). Note that both the normal and side mask supervisions are updated as the input of the HGM and the normal estimator also contains the updated SMPL-X in each iteration. We provide more analysis on our SMPL-X refinement in the appendix.

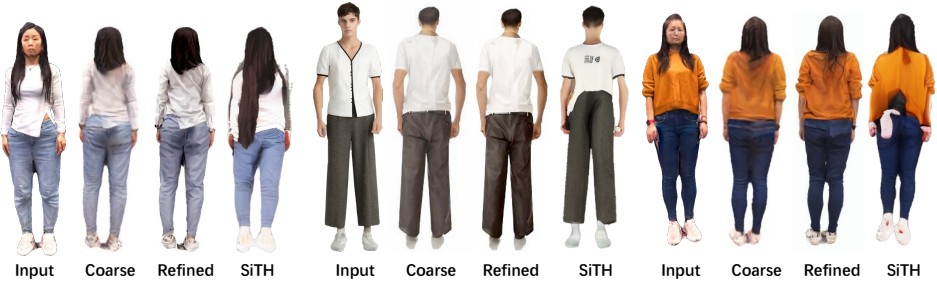

| Input | Coarse | Refined | SiTH | Input | Coarse | Refined | SiTH | Input | Coarse | Refined | SiTH |

Figure 4: Our back-view refinement can generate more realistic back-view images, compared with back-view hallucination of SiTH Ho et al. (2024).

## 3.4 BACK-VIEW REFINEMENT

Back-view hallucination poses a significant challenge in single-view human reconstruction. As shown in Fig. 4, directly using diffusion models to generate a back-view image can result in incorrect perspective projection with the front-view image as well as unrealistic texture by (Ho et al., 2024). The reason is that their diffusion is conditioned on the back-view mask, and thus can only be applied for orthogonal projection where back-view mask can be directly flipped with the front-view image. However, the back-view mask is not available during our inference stage since our prediction is based on perspective projection. To address this issue, we adopt a generate-then-refine strategy that leverages the diffusion prior to produce a perspective-fitted and realistic back-view image that is suitable for the subsequent two-view fusion stage. We train a ControlNet (Zhang et al., 2023b) to give realistic details based on our coarse results as shown in the right of Fig. 3. Specifically, we generate coarse back-view rendering by our HGM for the training dataset and only train the ControlNet and keep the base Stable Diffusion model as fixed. The ControlNet loss is given as:

$$\mathcal{L}_{CN} = \mathbb{E}_{\mathbf{z}_0, t, \mathbf{c}_t, \boldsymbol{\epsilon}_t \sim \mathcal{N}(0,1)} \left[ \| \boldsymbol{\epsilon} - \epsilon_\theta(\mathbf{z}_t, t, \mathbf{c}_t, \mathbf{y}) \|_2^2 \right], \tag{4}$$

where $y$ is the text prompt. We set it as 'Best quality' and the negative prompt as 'blur, bad anatomy, bad hands, cropped, worst quality' during inference. $\mathbf{c}_t$ is the coarse back-view image from our HGM rendering $I_B^c$, which is the ControlNet condition. We carefully design the reversing process by adding small amount of noise to the VAE encoded latent of $I_B^c$ to keep the original content as much as possible. The sampling process takes around 2 seconds. In Fig. 4, we show our generated and refined back-view results, comparing them with the back-view hallucination diffusion network in SiTH (Zhang et al., 2024). Our results maintain high resolution and generate details such as the hair of the first woman and the wrinkles in the clothes. In comparison, SiTH (Zhang et al., 2024) produces artifacts and unrealistic hallucination results. Moreover, the results are also not fitted perspectively to the input image.

## 3.5 TWO-VIEW RECONSTRUCTION

We combine the refined perspective-fitted back-view image $I_B^r$ with the front-view image $I$ and input them into the fusion HGM model to get:

$$G_{refine} = \text{HGM}(I, I_B^r). \tag{5}$$

Specifically, our fusion HGM model retains the design of the coarse HGM model with the additional refined back-view image as the input. The coarse HGM and fusion HGM models are trained separately with ground-truth one view and two views as input, as well as ground-truth SMPL-X. The objective function for HGM training includes $\mathcal{L}_2$ color loss, $\mathcal{L}_{rgb}$, VGG-based LPIPS perceptual loss, $\mathcal{L}_{lpips}$ (Zhang et al., 2018), and $\mathcal{L}_2$ background mask loss $\mathcal{L}_{bg}$ with ground truth masks. Each of these losses has corresponding weights that are treated as hyperparameters:

$$\mathcal{L}_{HGM} = \lambda_{rgb}\mathcal{L}_{rgb} + \lambda_{lpips}\mathcal{L}_{lpips} + \lambda_{bg}\mathcal{L}_{bg}, \tag{6}$$

where $\lambda_{rgb} = \lambda_{lpips} = \lambda_{bg}$=1.0. $\mathcal{L}_{HGM}$ is applied for coarse HGM and fusion HGM.

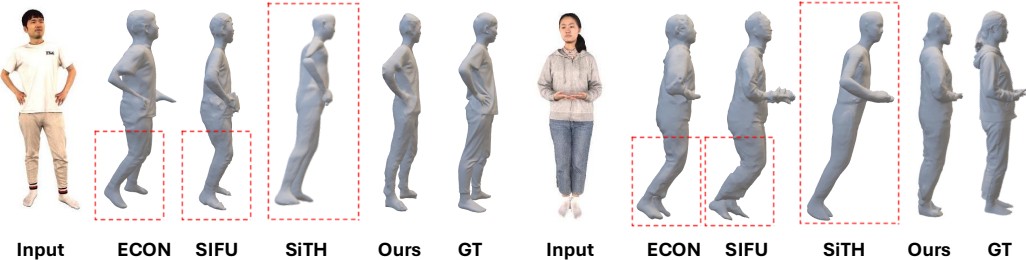

Figure 5: Levraging our HGM model, SMPL-X parameters are iteratively refined to mitigate the issue of blended legs commonly seen in other approaches.

## 3.6 IMPLEMENTATION

Our model is trained on 4 NVIDIA RTX A6000 with batch size of 4 for 20 hours. Our input image size is 512×512 and the number of Gaussians for each view is 256×256, with a total of 65,536 Gaussians per view. For SMPL-X estimation, we use PIXIE(Feng et al., 2021). Network structures and more implementation details are in the appendix.

## 4 EXPERIMENTS

We conduct experiments on the publicly available 3D human datasets THuman2.0 (Yu et al., 2021), CustomHumans(Ho et al., 2023) and HuMMan (Cai et al., 2022). Our method is compared with state-of-the-art (SOTA) methods in both novel view synthesis and 3D mesh reconstruction. We train our HGM on 500 human scans from THuman2.0 dataset following Zhang et al. (2024). We render the images with resolution of 512×512 and using weak perspective camera on 12 fixed cameras evenly distributed with the azimuths from 0 to 360 degree. During evaluation, all the methods are tested without the ground truth SMPL-X. We follow the train and test list from SIFU (Zhang et al., 2024) and SHERF (Hu et al., 2023) to evaluate our method on THuman2.0 and HuMMan dataset. For CustomHumans dataset we use 45 scans for cross-dataset evaluation containing loosing clothes and challenging poses. For novel view synthesis, We use PSNR, SSIM, LPIPS as evaluation metrics. For 3D reconstruction, we use commonly used Chamfer Distance (CD), Point-to-Surface Distance (P2S), and Normal Consistency as the evaluation metrics.

## 4.1 NOVEL VIEW SYNTHESIS

Table 1: Novel view synthesis comparison with SOTA methods.

| Method | THuman2.0 | | | CustomHumans | | |
|---|---|---|---|---|---|---|
| | PSNR ↑ | SSIM ↑ | LPIPS ↓ | PSNR ↑ | SSIM ↑ | LPIPS ↓ |
| GTA Zhang et al. (2023c) | 19.09 | 0.882 | 0.113 | 19.59 | 0.887 | 0.125 |
| SiTH Ho et al. (2024) | 17.12 | 0.843 | 0.155 | 18.09 | 0.856 | 0.144 |
| LGM Tang et al. (2024a) | 18.34 | 0.856 | 0.134 | 19.87 | 0.877 | 0.132 |
| SV3D Voleti et al. (2024) | 19.11 | 0.892 | 0.117 | 20.86 | 0.902 | 0.112 |
| SIFU Zhang et al. (2024) | 22.10 | 0.924 | 0.0794 | 20.83 | 0.898 | 0.117 |
| **Ours** | **23.54** | **0.938** | **0.0524** | **23.84** | **0.944** | **0.0514** |

For novel view synthesis, we compare our method with mesh SOTA human reconstruction methods GTA Zhang et al. (2023c), ECON Xiu et al. (2023), SIFU Zhang et al. (2024) and SiTH (Ho et al., 2024), as well as multiview diffusion reconstruction method LGM(finetuned with the same training data) Tang et al. (2024a) and video diffusion method SV3D Voleti et al. (2024) on THuman2.0 Yu et al. (2021) and CustomHuman Ho et al. (2023). We also compare our method with state-of-the-art HumanNeRF methods: SHERF (Hu et al., 2023), MPS-NeRF (Gao et al., 2022), and NHP (Kwon et al., 2021) on the HuMMan dataset (Cai et al., 2022).

As shown in the Tab. 1 and Tab. 2, our method significantly surpasses state-of-the-art single-view human reconstruction methods in all evaluation metrics for the three datasets. As shown in Fig. 6, LGM Tang et al. (2024a) generates incorrect blue color and inconsistent content. SiTH Ho et al. (2024) fails to model loose clothes due to the high dependency of the SMPL-X model. Side views are blurry and unrealistic in SIFU's results. SV3D Voleti et al. (2024) generates strange colors and wrong human pose. Compared with these methods, ours generates more realistic and consistent rendering especially for the unseen regions such as clothes wrinkles and hair that are well-fitted to the front views and more robust to initial SMPL-X estimation errors thanks to our iterative refinement. We provide rendering videos in 360 degree comparison with other methods in the Appendix. We also provide a visual comparison with the SOTA NeRF-based method SHERF Hu et al. (2023) on the HuMMan dataset in the appendix.

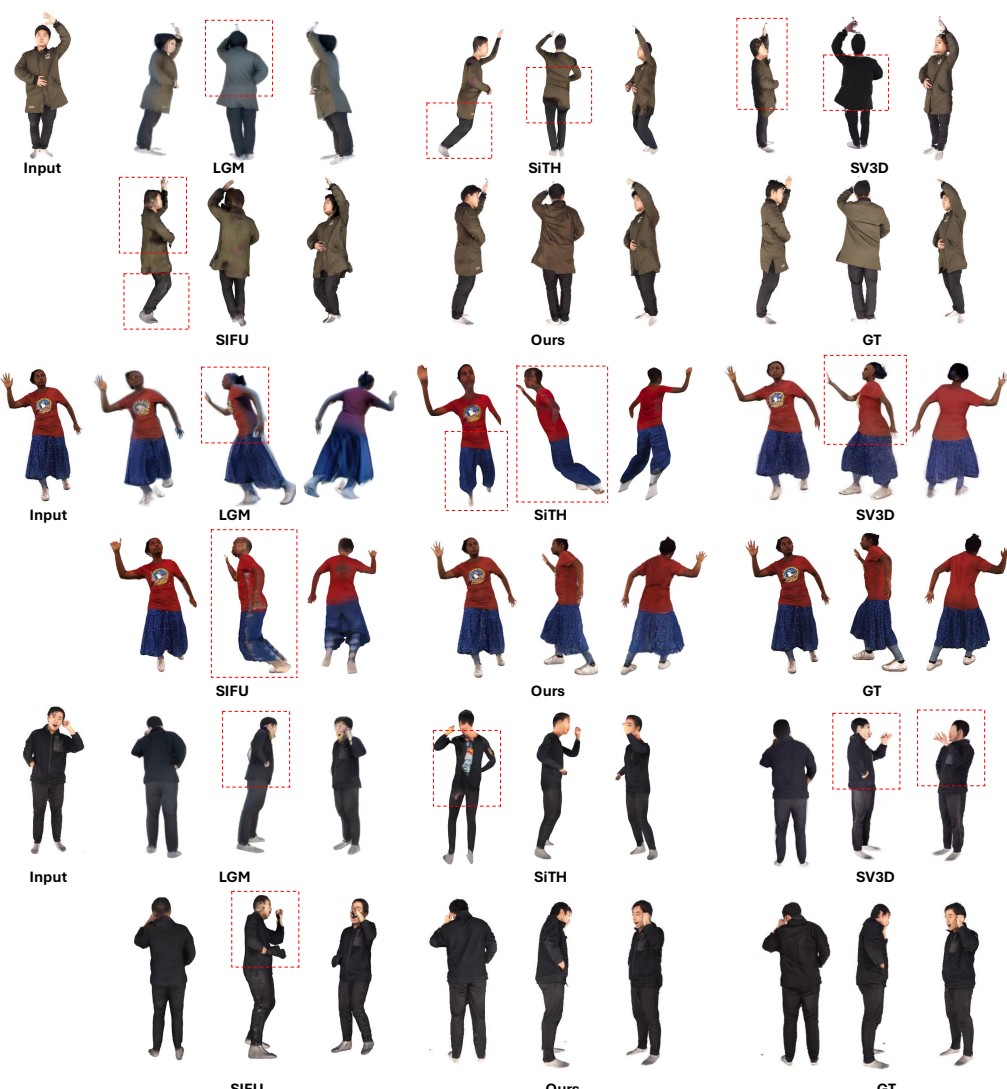

Figure 6: Novel view synthesis comparison with other approaches on THuman2.0 and CustomHumans dataset. **The details are highlighted in the red boxes.**

## 4.2 3D RECONSTRUCTION

For mesh reconstruction we extract the 3D mesh by densely rendering the depth map with Gaussian render and using TSDFusion to extract the surface followed by a fast optimizaiton based on the normal map obtained in section3.3.2. We compare our results with SOTA human surface reconstruction methods GTA (Zhang et al., 2023c), ECON (Xiu et al., 2023), SIFU (Zhang et al., 2024) and SiTH

Table 2: Novel view synthesis comparison with SOTA HumanNeRF methods on HuMMan.

| Method | PSNR ↑ | SSIM ↑ | LPIPS ↓ |
|---|---|---|---|
| NHP (Kwon et al., 2021) | 18.99 | 0.845 | 0.182 |
| MPS-NERF (Gao et al., 2022) | 17.44 | 0.824 | 0.193 |
| SHERF (Hu et al., 2023) | 20.83 | 0.891 | 0.125 |
| Ours | **23.86** | **0.952** | **0.0591** |

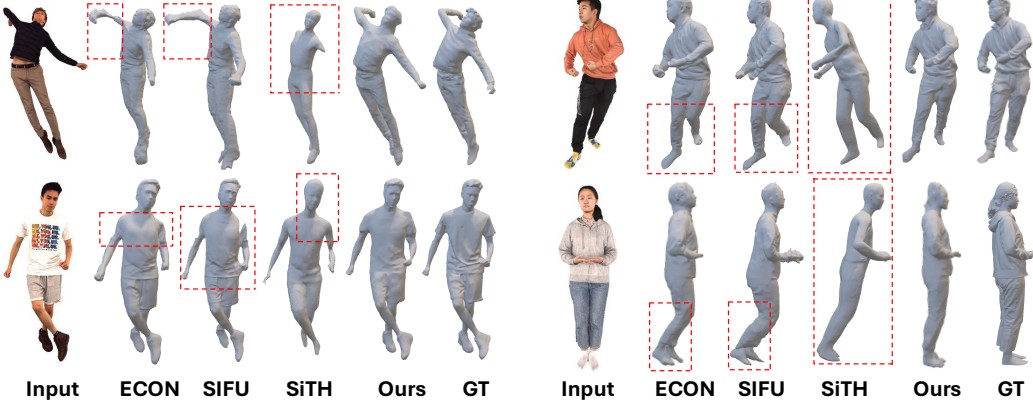

Figure 7: 3D reconstruction visualization compared with SOTA methods. **Details are highlighted in the red boxes.**

Ho et al. (2024). Note that our method do not use the 3D ground truth for supervision, but can also achieve best performance compared with all those fully supervised methods. Thanks to our interative SMPL-X refinement. Our method alleviates commonly occurring problems of bent legs and incorrect postures found in previous methods, as shown in Fig.5. Our HGM model reconstructs more accurate geometry with the prior learned from our dual branch Gaussian reconstruction model as well as our innovative SMPL-X refinement. We provide a qualitative 3D reconstruction comparison in Fig.7. As shown in the figure, ECON and SIFU suffer from bending legs and wrong arms problems. SiTH generates an over-smoothed surface and missing parts. Our method can reconstruct more accurate poses while preserving geometric details.

Table 3: 3D reconstruction comparison with SOTA methods. Note that only our method is trained *without* 3D supervision.

| Method | THuman2.0 | | | CustomHumans | | |
|---|---|---|---|---|---|---|
| | Chamfer ↓ | P2S ↓ | Normal ↑ | Chamfer ↓ | P2S ↓ | NC ↑ |
| ECON Xiu et al. (2023) | 2.342 | 2.431 | 0.765 | 2.107 | 2.355 | 0.771 |
| GTA Zhang et al. (2023c) | 2.201 | 2.314 | 0.773 | 1.987 | 2.115 | 0.769 |
| SiTH Ho et al. (2024) | 2.519 | 2.442 | 0.786 | 2.223 | 2.584 | 0.785 |
| SIFU Zhang et al. (2024) | **2.063** | 2.205 | 0.792 | 1.864 | 1.976 | 0.778 |
| Ours | 2.134 | **2.118** | **0.823** | **1.729** | **1.835** | **0.834** |

### 4.3 ABLATION STUDIES

We conduct ablation studies to evaluate the effectiveness of our SMPL-X dual branch Gaussian prediction model, coarse-to-fine refinement strategy, back-view refine ControlNet, and our SMPL-X refinement. We show the quantitative ablation results in Table 4. The performance decreases when any component is removed. SMPL-X dual branch plays an important role in adding human priors through structural features to Image Gaussians predicted by UNet. We visualize the rendered images using our model without SMPL-X dual branch as the guidance and those produced by our full model, as shown in Fig. 8. Without SMPL-X dual branch as guidance, the side-view images exhibit significant artifacts,

such as misaligned arms and unnatural shapes of clothes and heads, highlighted in the red boxes. This demonstrates the effectiveness of our SMPL-X dual branch Gaussian prediction design. The predicted Gaussians lack human shape and pose priors without SMPL-X guidance, resulting in unnatural shapes and poses. Since the initial SMPL-X prediction is not accurate, we ablate the effectiveness of our iterative SMPL-X refinement on the 3D prior learned in our HGM model. We also visualize the SMPL-X refinement in the appendix. Two-view refinement doubles the number of Gaussians to improve the reconstruction quality. Gaussians tend to concentrate more on the front view without the two-view refinement strategy, leading to poorer rendering of the back part. Additionally, the diffusion-based refinement is crucial for improving the novel view synthesis quality, especially for the back-view images as shown in Fig. 4. The best performance is achieved with all three components.

Table 4: Ablation study for each component.

| Components | NVS | | | 3D reconstruction | | |
|---|---|---|---|---|---|---|
| | PSNR(↑) | SSIM(↑) | LPIPS(↓) | Chamfer(↓) | P2S(↓) | NC(↑) |
| w/o SMPL-X dual branch | 21.85 | 0.908 | 0769 | 2.421 | 2.543 | 0.776 |
| w/o SMPL-X refine | 22.86 | 0.921 | 0.656 | 2.245 | 2.346 | 0.798 |
| w/o two-view refine | 22.95 | 0.924 | 0.641 | 2.301 | 2.343 | 0.781 |
| w/o Diffusion refine | 23.11 | 0.921 | 0.637 | 2.145 | 2.176 | 0.812 |
| Full model | **23.54** | **0.938** | **0.524** | **2.134** | **2.118** | **0.823** |

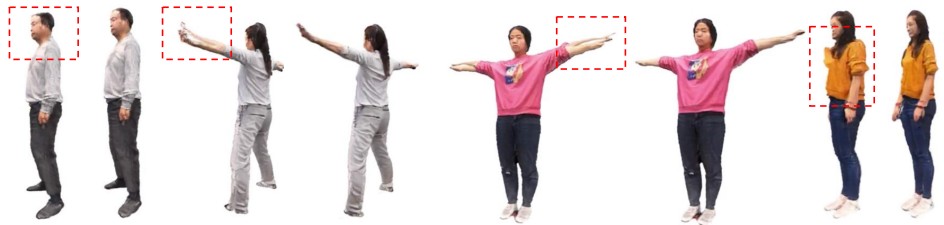

Figure 8: Ablation studies in terms of SMPL-X dual branch guidance. For each pair of images, left one is the results of our model w/o SMPL-X guidance. **Details are highlighted in the red boxes.**

## 4.4 DISCUSSIONS

Our method improves upon previous approaches like ECON Xiu et al. (2023) and SIFU Zhang et al. (2024) by leveraging our pre-trained HGM model to incorporate 3D priors, specifically side view masks, for enhanced SMPL-X refinement. The refined SMPL-X also benefits our human Gaussians reconstruction by providing structural information encoded in learnable tokens. Our 3D Gaussians-based method offers significant advantages in rendering speed, achieving 300 FPS compared to NeRF-based methods like SHERF Hu et al. (2023), which manages only 2 FPS. Furthermore, the mesh extracted from our 3D Gaussians with normal refinement attains high 3D reconstruction quality. In summary, our method excels in both high-quality rendering and accurate 3D reconstruction, offering a comprehensive solution.

## 5 CONCLUSION

In this paper, we introduce a novel generalizable single-view human Gaussian reconstruction framework. By incorporating human priors through a SMPL-X dual branch Gaussian prediction and diffusion priors using a refinement ControlNet, our method effectively handles invisible parts and varying poses. By incorporating our pre-trained HGM, inaccurate SMPL-X is iteratively refined, which benefits the Gaussian reconstruction quality. Combining all of these techniques, our method can generalize well to unseen subjects for high-quality and view-consistent reconstruction. We validate the proposed method on several benchmarks and demonstrate that it achieves state-of-the-art performance in both novel view synthesis and 3D reconstruction.

**Acknowledgement.** This research / project is supported by the National Research Foundation (NRF) Singapore, under its NRF-Investigatorship Programme (Award ID. NRF-NRFI09-0008).

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

## A APPENDIX

We introduce the following content in the appendix: SMPL-X optimization and mesh optimization details, SMPL-X evaluation, back-view details and evaluation, additional comparison, experimental environment, network structures, limitations, and more visualizations.

**SMPL-X optimization and mesh optimization details.**   During optimization, we render SMPL-X side views and compute the side-view mask loss and normal loss for a total of 45 iterations. SMPL-X parameters are updated at each iteration. The updated SMPL-X parameters are fed into HGM to update GS every 15 iterations (3 times in total), reducing the overall optimization time. The initial SMPL-X are not input to HGM for GS prediction until after the first 15 iterations because poorly-aligned SMPL-X can lead to degraded GS. For the front view, we utilize the original image mask instead of the one rendered from GS to stabilize the process. We simultaneously render 12 views (one front-view and all the other views are considered as side views) to compute the mask loss. The loss weights are set as follows: $\lambda_{front} = 10$, $\lambda_{side} = 1$, and $\lambda_n = 0.5$. For the normal loss, we only utilize the front and back views with a pre-trained normal estimator from ICON. Throughout the optimization process, HGM remains fixed while only SMPL-X parameters are updated. The elegance of our method lies in its iterative nature: GS refines SMPL-X and better-aligned SMPL-X estimates feed back into the HGM model to generate improved 3D Gaussians, which in turn enhance the reconstruction. We show the visualization of our side-view mask rendered from iteratively reconstructed Gaussisnas by HGM, initial SMPL-X, refined SMPL-X, and our finial-extracted meshes in Fig. 11. As shown in the figure, the side view masks effectively help refine the initial SMPL-X error for accurate reconstruction. For mesh refinement, we minimize the $\mathcal{L}_1$ loss between the predicted normal map and the rendered normal map. We also add Laplacian loss for the preservation of the local structure.

**Additional comparison with TeCH.**   We compare our method with TeCH on the Customhumans dataset quantitatively in Tab. 5. TeCH needs 4-5 hours for each sample, so we use 10 samples from the CustomHumans dataset for comparison. TeCH Huang et al. (2024) has several obvious limitations compared with ours: 1) The geometry refinement from SDS is not stable and the surface is broken as shown in the left part of Fig 9 even though capturing more high-frequency geometric details. 2) Slow optimization: It needs 4-5 hours optimization, while ours use only around 90s. 3) The caption guidance can sometimes be incorrect. For example, as shown in the left part of Fig 9, the wrong caption of the gender resulted in wrong face reconstruction. 4) SMPL-X error leading to bending legs and wrong geometry, which is the same issue in SIFU, SiTH, ECON and GTA as shown in SIFU, SiTH, ECON and GTA.

Table 5: Additional evaluation with TeCH.

| | PSNR(↑) | SSIM(↑) | LPIPS(↓) | CD(↓) | P2S(↓) | NC(↑) |
|---|---|---|---|---|---|---|
| Ours | 24.56 | 0.949 | 0.051 | 1.715 | 1.844 | 0.833 |
| TeCH | 23.87 | 0.927 | 0.079 | 2.232 | 2.432 | 0.778 |

**SMPL-X refinement evaluation.**   We evaluate using SMPL-X initializations from PyMAF-X Zhang et al. (2023a) and PIXIE Feng et al. (2021). We compute the MPJPE (mm) using the first 22 body joints defined in SMPL-X on both THuman2.0 Yu et al. (2021) and CustomHumans Ho et al. (2023) datasets. In the Tab. 6, 'Initial' means the direct estimation from SMPL-X predictors. 'w/o side-views' represents optimization without side-views mask loss. 'ours' refers to our optimization with all losses, including the side-views mask loss. We can see from the table that our method successfully refines the initial SMPL-X estimates using side-view priors from our HGM, which significantly reduces the error compared with without using side-view masks. Although PyMAF-X provides better initial SMPL-X estimates than PIXIE, both methods achieve comparable MPJPE scores after optimization, as the side-view mask loss guides them toward similar convergence points. This also shows that our method is robust to diverse SMPL-X initial estimators and can effectively improve the initial SMPL-X estimation.

**Backview refinement details and evaluation.**   We use the original ControlNet architecture and initialize the ControlNet with the ControlNet-tile model. ControlNet-tile is originally trained as an image super-resolution model. We finetune the ControlNet part with our constructed data pair at

Table 6: SMPL-X refinement evaluation in terms of MPJPE.

| Dataset | PIXIE | | | PyMAF-X | | |
|---|---|---|---|---|---|---|
| | Initial($\downarrow$) | w/o side-views($\downarrow$) | Ours($\downarrow$) | Initial($\downarrow$) | w/o side-views($\downarrow$) | Ours($\downarrow$) |
| CustomHumans | 75.79 | 65.33 | 39.11 | 65.20 | 58.12 | 39.78 |
| Thuman2.0 | 80.11 | 72.36 | 44.30 | 71.18 | 65.65 | 44.84 |

Figure 9: Visual comparison with TeCH on loose cloth and challenging pose cases.

learning rate of 1e-5, with the base SD1.5 keep fixed. Data pair construction involves first training our HGM using single-view input without full convergence. Subsequently, we perform inference, downsample and render the back-view. The resulting back-view renderings, intentionally designed to have lower quality, serve as conditioning inputs for our ControlNet training. To validate the effectiveness of our proposed back-view refinement strategy, we performance quantitative evaluation with SiTH Ho et al. (2024) and Huang et al. (2024) for back-view quality on the CustomHumans dataset. We use SSIM, LPIPS and KID as evaluation metrics between the ground truth and generated back view images. SiTH generates back-view using pure hallucination, which always generate unrealistic image as shown in Fig. 4 and Tab. 7. TeCH use SDS loss to optimize the back-view. However, the back view always fits to the wrong SMPL-X pose and imprecise text description, which leads to lower generation quality.

Table 7: Backview evaluation.

| | SSIM($\uparrow$) | LPIPS($\downarrow$) | KID($\times 10^{-3}$ $\downarrow$) |
|---|---|---|---|
| Ours | 0.949 | 0.079 | 9.26 |
| SiTH | 0.855 | 0.123 | 29.8 |
| TeCH | 0.876 | 0.118 | 20.3 |

**Experimental environment.** We conduct all the experiments on NVIDIA RTX A6000 GPU. The experimental environment is PyTorch 2.2.1 and CUDA 12.2.

**Network structures.** Our UNet model consists of 6 down blocks, 1 middle block and 5 up blocks, with an input image size of 512×512 and an output Gaussian feature map size of 256×256. We use 2 input views, resulting in a total of $256 \times 256 \times 4 = 131,072$ output Gaussians. Each block contains several residual layers and an optional down-sample or up-sample layer. For the last 3 down blocks, the middle block, and the first 3 up blocks, we insert cross-view self-attention layers after the residual layers. The final feature maps are processed by a $1 \times 1$ convolution layer to produce 14-channel pixel-wise Gaussian features. We adopt SiLU activation and group normalization for the UNet. Our $\text{Tr}_{mix}$ share the same structure, consisting of multiple self-attention blocks. Specifically, they each have 5 up blocks and 5 down blocks. The down-sample channels are [64, 128, 256, 512, 1024], and the up-sample channels are [1024, 512, 256, 128, 64]. The input dimensions for $\text{Tr}_{mix}$ is 256,

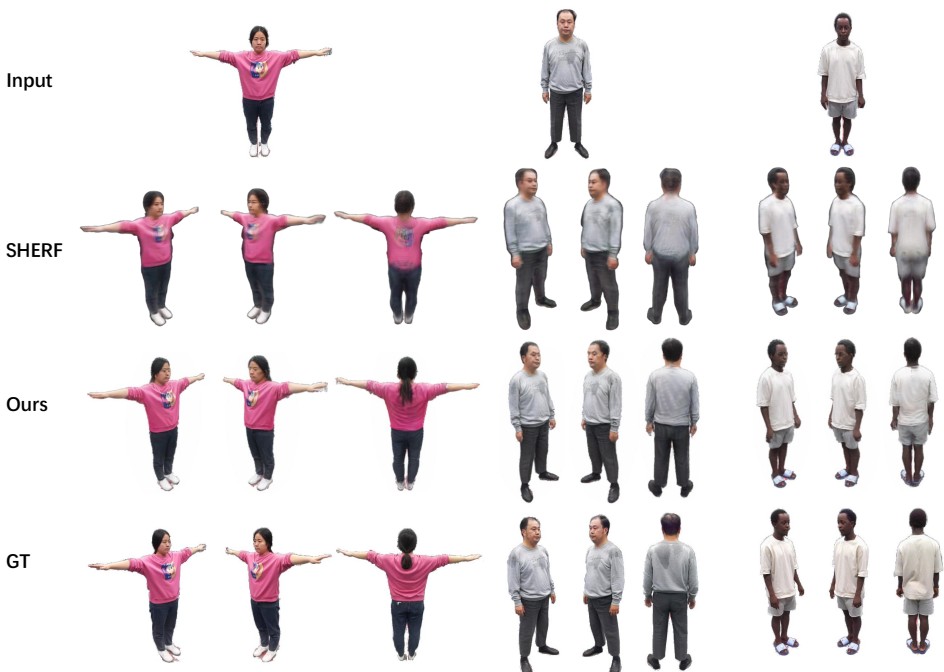

Figure 10: Novel view synthesis comparison SHERF on HuMMan dataset.

respectively. The output dimension for both is 14, which matches the dimension of the Gaussian features. For the attention blocks, we use a memory-efficient attention implementation.

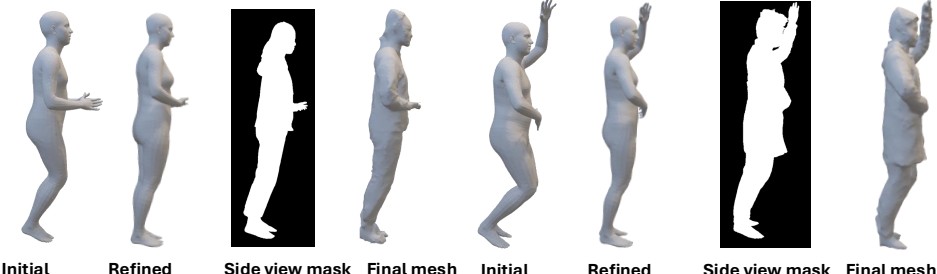

Figure 11: SMPL-X refinement visualization.

**Limitations.** Currently, our method is hard to generate high-quality hands and faces, which could potentially be solved by using the SMPL-X model and regional diffusion guidance such as SDS loss for further refinement. Also, the mesh extraction process from reconstructed gaussians is not straight forward, incurring additional optimization with estimated normal map as supervision.

**Additional results.** We give visual comparison with the SOTA NeRF-based method: SHERF Hu et al. (2023) shown in 10. While SHERF predicts blurry results and loses fidelity, our method preserves high-frequency details and generates realistic back views such as wrinkles and hair that fit well to the front views.

