# OpenReview forum: "Generalizable Human Gaussians from Single-View Image"
_ICLR.cc/2025/Conference — ICLR 2025 Poster_

### Official Review · Reviewer_e633 · 2024-10-30

**Soundness:** 4
**Presentation:** 3
**Contribution:** 3
**Rating:** 6
**Confidence:** 5

**Summary:**

This paper presents a generalizable single-view 3D human reconstruction system, which uses 3D Gaussians as 3D representations. Although the 3D Gaussian representation is promising due to its rendering speed, naively adapting 3D Gaussians in 3D human reconstruction often produces errored reconstructions. To resolve this issue, this paper introduces generate-then-refine pipeline that 1) reconstructs 3D Gaussians, 2) refines SMPL-X & back-view, and 3) reruns the 3D Gaussian reconstruction. The experiments demonstrate that it achieves state-of-the-art in novel-view synthesis and 3D reconstruction compared to existing methods.

**Strengths:**

In recent years, 3D Gaussian splatting (3DGS) has shown impressive results in 3D vision fields. Despite the promising results of 3DGS, the exploration of adapting 3D Gaussians for single-view 3D human reconstruction is still lacking. This paper points out the challenges of adapting 3D Gaussians for 3D human reconstruction. I think solving the proposed generate-then-refine pipeline could be one of the good research directions in 3D human reconstruction. The proposed approach sounds plausible, and can be plugged into most reconstruction algorithms. The paper is written in an easy-to-understand manner.

**Weaknesses:**

I have several concern points below.

1) About SMPL-X refinement strategy

It is unclear why the proposed SMPL-X refinement pipeline actually helps. As described in Section 3.3.2, the SMPL-X is optimized by two-type losses: normal loss and rendering loss. The optimization targets of two losses can be inaccurate: normal GT is from normal estimator, rendering T is from coarse Gaussians. Furthermore, as the coarse Gaussians are processed based on the initial SMPL-X (Figure 2), it seems that coarse Gaussians will not deviate much from the initial SMPL-X. It is questionable whether the proposed SMPL-X improvements always guarantee improved reconstruction quality. For example, if using a distinguished SMPL-X estimator, the refinement pipeline can lead to performance degradation.

2) Validation of SMPL-X refinement

Although SMPL-X refinement is claimed to be an important contribution, there is no detailed experiment on it. Although Table 4 indirectly demonstrates this, more explicit experiments on SMPL-X improvements (measuring MPJPE for before refine. / after refine.) are required.  Additionally, to strengthen the effect of SMPL-X refinement, experiments on diverse SMPL-X initial estimator [1,2,3] are also needed.


3) Description about ControlNet in back-view refinement

The paper does not specified how ControlNet in back-view refinement is trained. Back-view refinement is also a major contribution of this paper, so it is important to clarify the training procedure for the reproducibility. I would recommend describing ControlNet's architecture, learning strategy, and training dataset. Specifically, I curious about how to augment coarse image (ControlNet condition) for the training. Furthermore, I also recommend that conducting a quantitative comparison with SiTH's back-view estimation (Figure 4) to justify the proposed refinement module.

4) Writing

a. In Section 3.3.2, it is unclear whether SMPL-X only optimizes parameters or whether SMPL-X and HGM are jointly optimized.

b. In L123, it is an overclaim that using multi-view images is main strength over existing methods. This algorithm setup using 12 rendered multi-view images (L350) is not significantly different from that using a 3D scan. Fundamentally, this algorithm also used 3D scans (L349), and did not show results when trained with 12 multi-view images for the existing methods.



[1] Moon et al., Accurate 3D Hand Pose Estimation for Whole-Body 3D Human Mesh Estimation, CVPRW 2022.

[2] Lin et al., One-stage 3d whole-body mesh recovery with component aware transformer, CVPR 2023.

[3] Zhang et al., PyMAF-X: Towards well-aligned full-body model regression from monocular images., IEEE 2023.


In conclusion, the proposed approach looks plausible and well-designed, but I am concerned that the proposed generate-then-refine strategy is actually effective. I think it could be a good academic paper if it includes a more clear explanation about SMPL-X refinement (1, 2) and back-view refinement (3), which are the core of the generate-then-refine strategy.

**Questions:**

1) According to the paper, it looks like back-view refinement is quite effective. If back-view refinement improves the final Gaussians, can the improved Gaussian be a good source for back-view refinement? In other words, I wonder if performance can continue to improve if processes 2) and 3) in Figure 2 are performed cyclically. This experiment would be a good way to demonstrate how much the paper's approach improves reconstruction performance.

---

### Official Review · Reviewer_Kuxz · 2024-10-31

**Soundness:** 3
**Presentation:** 3
**Contribution:** 2
**Rating:** 6
**Confidence:** 4

**Summary:**

This paper proposes a novel approach to reconstruct human models from a single view. It first uses a coarse HGM model to predict GS representation. Besides feature from image space like the splatter image paper, this HGM model additionally aligns features based on the fitted SMPL-X template via a dual path. Then, the SMPL-X parameters and GS prediction are iteratively refined to improve mutual alignment. Finally, a ControlNet model generates a back view, which, combined with the front view, is processed by a two-view HGM for the final reconstruction.

**Strengths:**

1. Feature Combination: The integration of pose-aware and pixel-space features improves the quality of the coarse GS prediction.
2. Enhanced Quality: This approach surpasses existing single-view human reconstruction methods in output quality.

**Weaknesses:**

1. Incremental Novelty: There are several papers on 3D reconstruction topic that have adopted diffusion model to synthesize more views as the first step. This paper improves similar idea by using stronger controlNet signal, i.e. the rendering of course GS. This idea, while effective, is not very inspiring to me.

2. SMPL-X Refinement Concerns: Although justified in ablation studies, the proposed iterative SMPL-X refinement is a EM-like approach, while the previous works always refine SMPL parameters regarding the original input image. The proposed iterative refinement may depend heavily on initial GS and SMPL-X accuracy, making it sensitive and prone to degenerate solutions. Also, the wording in Section 3.3.2 is a bit redundant and would benefit from refinement for clarity and conciseness.

**Questions:**

The patch feature $F'_I$ on line 236 is ambiguous. it is unclear if this is derived from DINO or a trained model.

---

### Official Review · Reviewer_ckon · 2024-11-03

**Soundness:** 3
**Presentation:** 3
**Contribution:** 2
**Rating:** 6
**Confidence:** 5

**Summary:**

This work learns 3D human Gaussians from a single image, aiming to recover detailed appearance and geometry, including unobserved regions.
The paper proposes a single-view generalizable framework that utilizes a generate-then-refine strategy guided by SMPL and diffusion priors.
To enhance realism and accuracy, the method refines the SMPL model and employs SMPL priors to improve initial coarse human Gaussian estimations using sparse convolution and attention mechanisms.

**Strengths:**

- The problem addressed is quite challenging, and the designed network framework is reasonable, although the process is somewhat lengthy.
- The overall writing of the paper is clear and coherent.

**Weaknesses:**

- The results are quite unsatisfactory, as there is a noticeable discrepancy between the synthesized face and the original image. While this issue is mentioned in the limitations section, Figure 9 suggests that the images represent two different individuals.
- The comparison with other methods is insufficient, including approaches like TeCH[1] and GTA[2].
- The qualitative results appear somewhat blurry, with some input images also seeming unclear, particularly in the video results.
- Minor issues:  change "in 2" to "in Eq. 2" in line 248; clearly indicate the correspondence between F_I and F_S in the figures; missing references such as [3].

[1] TeCH: Text-guided Reconstruction of Lifelike Clothed Humans

[2] Global-correlated 3ddecoupling transformer for clothed avatar reconstruction

[3] HumanSplat: Generalizable Single-Image Human Gaussian Splatting with Structure Priors

----
I may change my opinion depending on the authors' rebuttal and whether they can address my concerns.

**Questions:**

The inference time is not reported; could you provide the specific timings for each step?

---

### Meta-Review · Area_Chair_NSi3 · 2024-12-17

**Metareview:**

This paper tackles the challenging problem of single-view 3D human reconstruction using 3D Gaussian representations. The method effectively addresses common limitations in detail recovery and novel-view synthesis by a generate-then-refine framework leveraging SMPL-X and diffusion priors. The integration of SMPL refinement, ControlNet-based back-view generation, and sparse convolution enhances reconstruction quality. Though concerns about SMPL-X refinement validation and back-view training details were noted, the reviewers generally agree that the contributions are sound via several rounds of discussions. The revised appendix is clear, and the method achieves state-of-the-art results. With revisions to address specific weaknesses, this paper is a valuable contribution to the research community.

**Additional Comments On Reviewer Discussion:**

The major concerns are facial discrepancy in rendering and SMPL-X refinement strategy. To address the former, the authors provided additional results in the revision demonstrating the effectiveness of their face rendering approach. For the latter, the authors presented further validation and evaluation of the SMPL-X optimization process. These efforts were acknowledged and recognized by the reviewers.

---

### Decision · Program_Chairs · 2025-01-22

Accept (Poster)